# Long-Term Sex- and Genotype-Specific Effects of ^56^Fe Irradiation on Wild-Type and APPswe/PS1dE9 Transgenic Mice

**DOI:** 10.3390/ijms222413305

**Published:** 2021-12-10

**Authors:** Maren K. Schroeder, Bin Liu, Robert G. Hinshaw, Mi-Ae Park, Shuyan Wang, Shipra Dubey, Grace Geyu Liu, Qiaoqiao Shi, Peter Holton, Vladimir Reiser, Paul A. Jones, William Trigg, Marcelo F. Di Carli, Barbara J. Caldarone, Jacqueline P. Williams, M. Kerry O’Banion, Cynthia A. Lemere

**Affiliations:** 1Department of Neurology, Ann Romney Center for Neurologic Diseases, Brigham and Women’s Hospital, Boston, MA 02115, USA; marenkschroeder@gmail.com (M.K.S.); binliu19810305@gmail.com (B.L.); rhinshaw@mit.edu (R.G.H.); gel32@pitt.edu (G.G.L.); shiqiaoqiao@gmail.com (Q.S.); 2Departments of Neurology, Harvard Medical School, Boston, MA 02115, USA; 3Harvard-MIT Division of Health Sciences and Technology, Massachusetts Institute of Technology, Cambridge, MA 02129, USA; 4Departments of Radiology, Harvard Medical School, Boston, MA 02115, USA; Mi-Ae.Park@UTSouthwestern.edu (M.-A.P.); sdubey1@bwh.harvard.edu (S.D.); pholton1@bwh.harvard.edu (P.H.); mdicarli@bwh.harvard.edu (M.F.D.C.); 5Department of Radiology, Brigham and Women’s Hospital, Boston, MA 02115, USA; swang14@bics.bwh.harvard.edu; 6GE Healthcare, Life Sciences, 101 Carnegie Center, Princeton, NJ 08540, USA; vladimir.reiser@gmail.com; 7GE Healthcare, Pharmaceutical Diagnostics, Pollards Wood, Nightingales Lane, Chalfont St. Giles HP8 4SP, UK; pauljones@ge.com (P.A.J.); william.trigg@hotmail.co.uk (W.T.); 8Mouse Behavioral Core, Harvard Medical School, Boston, MA 02115, USA; barbara_caldarone@hms.harvard.edu; 9Department of Environmental Medicine, University of Rochester Medical Center, Rochester, NY 14642, USA; jackie_williams@urmc.rochester.edu; 10Department of Neuroscience, Del Monte Neuroscience Institute, University of Rochester Medical Center, Rochester, NY 14642, USA; kerry_obanion@urmc.rochester.edu

**Keywords:** ^56^Fe radiation, Alzheimer’s disease, sex differences

## Abstract

Space radiation presents a substantial threat to travel beyond Earth. Relatively low doses of high-energy particle radiation cause physiological and behavioral impairments in rodents and may pose risks to human spaceflight. There is evidence that ^56^Fe irradiation, a significant component of space radiation, may be more harmful to males than to females and worsen Alzheimer’s disease pathology in genetically vulnerable models. Yet, research on the long-term, sex- and genotype-specific effects of ^56^Fe irradiation is lacking. Here, we irradiated 4-month-old male and female, wild-type and Alzheimer’s-like APP/PS1 mice with 0, 0.10, or 0.50 Gy of ^56^Fe ions (1GeV/u). Mice underwent microPET scans before and 7.5 months after irradiation, a battery of behavioral tests at 11 months of age and were sacrificed for pathological and biochemical analyses at 12 months of age. ^56^Fe irradiation worsened amyloid-beta (Aβ) pathology, gliosis, neuroinflammation and spatial memory, but improved motor coordination, in male transgenic mice and worsened fear memory in wild-type males. Although sham-irradiated female APP/PS1 mice had more cerebral Aβ and gliosis than sham-irradiated male transgenics, female mice of both genotypes were relatively spared from radiation effects 8 months later. These results provide evidence for sex-specific, long-term CNS effects of space radiation.

## 1. Introduction

The National Aeronautics and Space Administration (NASA) is preparing for lunar base habitation and multi-year, crewed missions to Mars. Exposure to space radiation poses a substantial threat to humans on these voyages, and the long-term consequences of such radiation exposure remain poorly studied. Ionizing space radiation consists of electromagnetic radiation such as gamma rays and particle radiation such as protons and high-charge and -energy (HZE) ions, the latter of which can pass through spacecraft shielding to create showers of secondary radiation upon impact. HZE particles in particular cause a high density of ionizations as they pass through matter, which makes them especially damaging to biological tissue. It is estimated that in interplanetary space, each cell in an astronaut’s body will be traversed by a HZE particle every few months [1] and that, over a 2.5 year Mars mission, astronauts will be exposed to 1 Sievert of space radiation [1,2]. According to the United Nations Scientific Committee on the Effects of Atomic Radiation (UNSCEAR) reports, the worldwide average effective dose per capita on Earth is 2.4 mSv/year from natural sources of radiation (UNSCEAR 2000 Report Vol.1). HZE particles are known to affect a variety of biological systems; previous research in rodents suggests that several months following low-dose (less than 1 Gy) ^56^Fe exposure, there is an increase in fibrosis-associated genes and hypermethylation in the lung [3] as well as aberrant blood chemistry associated with liver and kidney function [4]. While central nervous system (CNS) tissues have traditionally been considered radioresistant, increasing evidence suggests that low doses of HZE particle radiation pose a threat to these tissues as well. Thus, the potential long-term effects of HZE particle exposure on the CNS are of high importance to space medicine research.

^56^Fe particles are the single largest contributor to the biologically effective dose received in the unshielded space radiation environment [5] and may negatively affect CNS function in both the short and the long term. Previous research in rodents suggests that ^56^Fe exposure causes immediate neurochemical changes in the hippocampus and frontal cortex [6], impairs neurogenesis [7,8], upregulates apoptosis-regulating genes in the hippocampus [9], and causes genomic instability in the brain [10] in the days and weeks following irradiation. Doses as low as 0.50 Gy may impair neurogenesis for at least 3 months post-irradiation, including significant decreases in newly born neurons, astrocytes, and oligodendrocytes in the hippocampus, and increases in newly born microglia [11].

These ^56^Fe-induced biochemical and genomic changes in the CNS impact cognitive function in rodent models. Mice exposed to 0.5, 1.0, or 2.0 Gy and rats given 1.5 Gy of ^56^Fe radiation have shown memory impairment one month post-irradiation [12,13]. At 3 months post-irradiation, rats given doses as low as 0.20 Gy showed impairment on the Barnes maze [14,15], and attention shifting tasks [16,17]. ^56^Fe irradiation also caused impairments in the cued and contextual fear conditioning test in mice [18,19]. Therefore, there is reason to believe that radiation may pose risks to the cognitive health of astronauts. However, only a few studies have examined the long-term, cognitive impacts of ^56^Fe irradiation in rodents that persist after several months. We have previously shown that male mice receiving 0.10 Gy showed impairment on the contextual fear conditioning test 6 months following ^56^Fe irradiation [20], and only a handful of studies have investigated the persistence or incidence of radiation-induced cognitive dysfunction beyond 9 months. Low-dose HZE-irradiated animals exhibit radiation-induced changes in operant conditioning [21], fear extinction [22], and several other forms of memory function [22,23] over one year after exposure.

There is increasing evidence that the long-term effects of ^56^Fe radiation may interact with sex and genetic risk factors for neurodegenerative diseases [23], at least in part due to established links between radiation and excessive reactive oxygen species (ROS) and between ROS and neurodegenerative diseases [24,25]. We have previously shown that male mice exposed to 0.10 Gy showed early signs of memory impairment on the contextual fear conditioning test, while female mice did not [26]. Several studies examining the effect of Apolipoprotein E (ApoE) isoforms, a well-established genetic risk factor for Alzheimer’s disease (AD), on the response to ^56^Fe irradiation have suggested that radiation exacerbates cognitive deficits in transgenic mice [23,27,28,29,30]. In addition to ApoE, mutations in the amyloid precursor protein (APP) and presenilin 1 (PS1) have been linked to AD in humans [31]. In transgenic mice with APP mutations, irradiation reduces synaptic excitability in the hippocampi and causes increased accumulation of Ab plaques in transgenic mice with APP mutations [32]. Moreover, sex may interact with genotype effects to mediate the response to radiation in both mouse models and in humans. Current NASA radiation exposure limits are lower for females than for males due to epidemiological cancer risk [33]. Despite this, there is evidence that females are protected from X-ray, ^28^Si-, and mixed ion radiation-induced behavioral deficits, possibly due to the neuroprotective effects of estrogen [34,35,36]. However, studies using doses of greater than 1 Gy of gamma and ^56^Fe irradiation suggest that females may be more susceptible to behavioral deficits [18,23,27], though these higher doses are less relevant to a multi-year spaceflight mission.

In the present study, we exposed 4-month-old male and female, wild-type and APPswe/PS1dE9 mice to either 0, 0.10, or 0.50 Gy of whole-body ^56^Fe irradiation. At 11 months of age, we subjected the mice to a battery of cognitive tests and brain analyses to elucidate the long-term sex- and genotype-dependent behavioral, pathological and biochemical effects of low-dose ^56^Fe irradiation.

## 2. Results

### 2.1. ^56^Fe Irradiation Decreased the Survival Rate of APP/PS1 Females but Did Not Affect Other Health Measures

APP/PS1 females in the 0.50 Gy dose group had significantly lower survival rates than sham-irradiated APP/PS1 females; however, irradiation did not affect the survival rates of any other groups (Appendix A). Irradiation with ^56^Fe did not affect body weight, although all groups of mice were significantly heavier at 12 months of age than they were at 4 months due to aging (Appendix A). Irradiation with ^56^Fe did not affect general health as measured by the SmithKline Beecham, Harwell, Imperial College, Royal London Hospital, phenotype assessment (SHIRPA) test (Appendix A).

### 2.2. ^56^Fe Irradiation Caused More Behavioral Changes in Male Than in Female Mice

We used the open field (OF), Y maze, elevated plus maze (EPM), tail suspension (TST), contextual fear conditioning (CFC), rotarod, grip strength, wire hanging, startle, and prepulse inhibition (PPI) tests to measure ^56^Fe irradiation-induced behavioral changes. There were no differences between groups on grip strength (Appendix A), though there was a trend towards sham-irradiated WT females having higher latencies to fall in the wire hanging test than sham-irradiated APP/PS1 females, suggesting better endurance in WT female mice (Appendix A). In female mice, irradiation did not affect motor coordination, (Figure 1I), motor learning (Figure 1J), fear memory (Figure 1G) or spatial memory (Figure 1H), regardless of genotype. However, genotype did affect locomotor activity: sham-irradiated APP/PS1 mice traveled significantly greater distances in the OF test (Figure 1A, *p* = 0.0323, *p* = 0.0046 for females and males, respectively) and had more vertical counts (Figure 1C, *p* = 0.0025, *p* = 0.0021 females and males, respectively) than sham-irradiated WT mice.

Exposure to 0.50 Gy ^56^Fe irradiation increased locomotor activity of APP/PS1 female mice in the open field compared to 0.10 Gy (*p* = 0.0002) and sham-irradiation (Figure 1A, *p* = 0.0003). Conversely, 0.10 Gy irradiation decreased the number of arm entries of APP/PS1 males in the Y maze, suggesting reduced locomotion (Figure 1B, *p* = 0.0316). Similarly, both 0.10 and 0.50 Gy irradiation decreased rearing of male APP/PS1 mice in the OF (Figure 1C, *p* = 0.0398, sham vs. 0.50 Gy *p* = 0.0356). Interestingly, WT females receiving 0.50 Gy irradiation reared significantly more than sham-irradiated WT females (Figure 1C, *p* = 0.0304), but otherwise irradiation did not affect the performance of female WT mice on behavioral measures.

^56^Fe irradiation had little to no effects on anxiety-like behaviors, as measured by the percent open arm entries in the EPM (Figure 1E) or the percent distance traveled in the center of the open field (Figure 1F). Exposure to 0.50 Gy ^56^Fe impaired fear memory of WT males as measured by the CFC test (Figure 1G, *p* = 0.0282). There were also strong trends towards irradiated APP/PS1 males having impaired spatial memory on the Y maze test (Figure 1H). Female mice did not show irradiation-induced deficits in fear memory, spatial memory, or changes in anxiety-like behavior, regardless of genotype. However, 0.10 Gy irradiation decreased the time immobile of APP/PS1 females in the TST (Figure 1D, *p =* 0.0231), indicating a decrease in depressive-like behavior.

On the rotarod test, irradiation improved motor coordination in APP/PS1 males (Figure 1I) without affecting body weight (Appendix A). APP/PS1 males in the 0.50 Gy group had significantly higher latencies to fall than sham-irradiated APP/PS1 males (Figure 1I, *p* = 0.0008). Sham-irradiated WT males had significantly higher latencies to fall compared to sham-irradiated APP/PS1 males (*p* = 0.0278), suggesting that irradiation may rescue a genotype deficit in APP/PS1 mice (Figure 1I). However, there were no differences between groups on percent improvement on the rotarod test, suggesting that irradiation did not affect motor learning (Figure 1J).

There were a few irradiation-induced differences on the startle test (Appendix A). ^56^Fe-irradiated female APP/PS1 mice startled significantly more and male WT mice startled less at louder tones. Female APP/PS1 mice in the 0.50 Gy group startled significantly more to the 110 dB stimulus than female APP/PS1 mice in the sham and 0.10 Gy groups (*p* = 0.0412 and *p* = 0.0461, respectively). Female APP/PS1 mice in the 0.50 Gy group startled significantly more to the 120 dB stimulus than sham-irradiated APP/PS1 females (*p* = 0.0307). In contrast, male WT mice irradiated with 0.50 Gy ^56^Fe startled less to a 90 dB stimulus than male sham-irradiated WT males (*p* = 0.0215) while male WT mice in the 0.50 Gy group startled more to the 120 dB stimulus than WT males in the 0.10 Gy group (*p* = 0.0063). There were no radiation-specific effects on startle response in female WT or male APP/PS1 mice.

Despite the irradiation-induced differences between female groups on the startle test, there were no differences between female groups in the PPI test (Appendix A). However, at PPI74 and PPI82 male APP/PS1 mice in the 0.50 Gy group had significantly higher percentages of inhibition than male APP/PS1 mice in the 0.10 Gy group (*p* = 0.0429 and *p* = 0.0471, respectively).

### 2.3. ^56^Fe Irradiation Increased Neuroinflammation and Altered Plasma Cytokine Levels

We used microPET imaging to measure hippocampal uptake of ^18^F-GE180, a TSPO (18 kD Translocator Protein) ligand PET tracer, as one measure of neuroinflammation 2 weeks before irradiation at 3.5 months of age and again 7.5 months after irradiation at 11.5 months of age in a subset of mice. The upregulation of TSPO is correlated with neuroinflammation. Thus, visualization of TSPO via PET imaging allows visualization of neuroinflammation. There were no sex or genotype differences in baseline hippocampal uptake of ^18^F-GE180 (Figure 2C,D). Sham-irradiated male and female WT mice showed small increases in uptake, likely due to aging (Figure 2E,F). Both sham-irradiated female and male APP/PS1 mice showed increases in uptake, likely due to aging and accumulation of Aβ pathology (Figure 2G,H). All mice showed increased tracer uptake following high-dose irradiation, but APP/PS1 mice of both sexes showed increased uptake compared to WT mice of both sexes (Figure 2I–L).

We normalized the post-irradiation ^18^F-GE180 PET signal by the pre-irradiation ^18^F-GE180 signal for each group and then compared the ratios of post/pre-PET signal between sham-irradiation and 0.50 Gy of ^56^Fe irradiation to investigate the effects of irradiation on neuroinflammation (Figure 3). The normalized data showed that the APP/PS1 males that received 0.50 Gy of irradiation had significantly higher ratios of post/pre-tracer uptake than the ratios of post/pre-tracer uptake in sham-irradiated APP/PS1 males (Figure 3D, *p* = 0.0117), but there were no differences in uptake before and after irradiation in WT males or in females with or without irradiation (Figure 3A–C). This suggests that irradiation causes increased neuroinflammation specifically in APP/PS1 male mice.

Moreover, we measured plasma cytokines and found that exposure to 0.10 Gy ^56^Fe, compared to 0.50 Gy, decreased plasma IL-10, an anti-inflammatory cytokine, in APP/PS1 males (Figure 4A, *p* = 0.0024). Exposure to 0.10 Gy irradiation significantly decreased plasma IFNγ in WT females (*p* = 0.0269), while sham-irradiated APP/PS1 females had significantly more plasma IFNγ than APP/PS1 males (Figure 4B, *p* = 0.0241). APP/PS1 females also had significantly more plasma IL-4 than APP/PS1 males (*p* = 0.0001), though 0.10 Gy irradiation significantly decreased IL-4 in APP/PS1 females (Figure 4E, *p* = 0.0341). Exposure to 0.50 Gy irradiation also significantly increased plasma TNFα compared to 0.10 Gy in APP/PS1 females (Figure 4J, *p* = 0.0317). In APP/PS1 males, 0.50 Gy irradiation increased IL-6 compared to 0.10 Gy. We did not find baseline genotype effects on plasma cytokine levels except that sham-irradiated APP/PS1 females had significantly higher levels of plasma KC-GRO than sham-irradiated WT females (Figure 4I, *p* = 0.0368). Little to no effects of irradiation, genotype, or sex were observed on plasma levels of IL-1β, IL-2, IL-5, or IL-12p70 (Figure 4C,D,F,H).

### 2.4. ^56^Fe Irradiation Increased Insoluble Aβ Load and Microgliosis in APP/PS1 Male Mice but Did Not Affect Synaptic Markers

We used the MSD Aβ-triplex ELISA (4G8) to quantify the long-term effects of ^56^Fe irradiation on cerebral Aβ levels. Sham-irradiated female APP/PS1 mice had significantly more insoluble Aβx-40 and insoluble Aβx-42 (Figure 5A,B, *p* < 0.0001 and *p* = 0.0002) than sham-irradiated APP/PS1 male mice, though 0.50 Gy ^56^Fe irradiation significantly increased insoluble Aβx-40 levels in APP/PS1 males (Figure 5B, *p* = 0.0018) but not in females. We also measured plaque load by immunolabeling brain slices with the anti-Aβ antibodies R1282 and 3A1 and β-sheet structured dye Thioflavin S, which stains for fibrillar Aβ. Sham-irradiated APP/PS1 females showed a significantly higher percent region of interest (ROI) of both 3A1 and R1282 hippocampal staining than sham-irradiated APP/PS1 males (*p* = 0.0205 and *p* = 0.0067, respectively), further supporting that APP/PS1 females had a greater plaque burden than APP/PS1 males (Figure 5C–E). Compared to 0.10 Gy ^56^Fe, 0.50 Gy increased plaque load as measured by R1282 in APP/PS1 males (Figure 5E, *p* = 0.0327). However, ^56^Fe irradiation did not affect plaque load in APP/PS1 females. Irradiation did not affect fibrillar Aβ as measured by percent ROI of Thioflavin S staining of the hippocampus (Figure 5F,G).

Gliosis was measured by Iba-1, CD68, and TSPO, markers for microglia, as well as by the astrocytic marker GFAP. There were irradiation and genotype, but not sex, effects on hippocampal Iba-1 staining. Sham-irradiated WT males and females had significantly lower Iba-1-positive staining than sham-irradiated APP/PS1 males and females, respectively (Figure 6A, *p* = 0.0358 and *p* = 0.012, respectively). Exposure to 0.50 Gy ^56^Fe irradiation significantly increased the percent ROI of Iba-1 staining in APP/PS1 males (*p* = 0.037), with a similar trend in APP/PS1 females (Figure 6A). APP/PS1 males and females were further examined for irradiation effects using other microglial markers. Sham-irradiated APP/PS1 females had significantly higher percent ROI of CD68 immunoreactivity than sham-irradiated APP/PS1 males (*p* = 0.0008), likely due to higher amyloid plaque load. Exposure to 0.50 Gy ^56^Fe irradiation produced a non-significant trend of increased CD68 staining in APP/PS1 males compared to sham APP/PS1 males (Figure 6B). There were no significant differences between groups on TSPO staining (Figure 6C). Exposure to 0.10 Gy ^56^Fe irradiation increased hippocampal astrocytic staining in APP/PS1 females compared to sham-irradiated APP/PS1 females (*p* = 0.0466) but not in APP/PS1 or WT males. Sham-irradiated APP/PS1 males showed significantly higher astrogliosis than sham-irradiated WT males, but no such differences were seen between sham-irradiated WT and APP/PS1 females (Figure 6D, *p* = 0.012). However, ^56^Fe had little to no effect on the number of microhemorrhages as detected by staining for hemosiderin deposits using Prussian blue (Figure 6E).

We used Western blots to measure whole-brain levels of presynaptic markers synaptophysin and VGluT2, as well as postsynaptic markers PSD95 and Homer1. There were no differences between groups on any of these markers (Appendix A). We also used immunofluorescence to determine the integrated density of synaptophysin, PSD95, and the dendritic marker MAP2 in the prefrontal cortex and in the CA1 and CA3 regions of the hippocampus. Irradiation did not change the integrated density of synaptophysin, PSD95, and MAP2 in any group, regardless of brain region (Appendix A).

## 3. Discussion

Heavy ion radiation encountered during spaceflight could cause significant sex-specific physiological and behavioral changes in astronauts, but the long-term consequences of these changes, and how they may interact with genetic predispositions to Alzheimer’s disease, are largely unknown. Many previous studies with ^56^Fe radiation have examined the short-term radiation effects in wild-type, male rodents. The present study is one of the first to investigate the long-term biochemical, pathological, and behavioral effects of ^56^Fe irradiation on male and female mice in the context of neurodegenerative disease predisposition. Overall, we demonstrated a general trend for male APP/PS1 mice to be more susceptible to detrimental radiation effects on pathology and cognitive function. Female APP/PS1 mice had elevated amyloid-related AD pathologies compared to APP/PS1 males but these pathologies were not exacerbated following irradiation. These data, showing radiation effects predominately in AD model mice over WT mice, demonstrate the importance of investigating additional stressors—in this case, AD pathology—and how they can unmask radiation-induced deficits that would otherwise be obscured by the individual’s varying capacity for compensation. Not only does incorporating other stressors reveal potentially unique interaction effects, it also adds fidelity to the multi-stressor environment of spaceflight.

We found that 0.50 Gy ^56^Fe radiation increased locomotor activity in female APP/PS1 mice, while 0.10 Gy ^56^Fe radiation decreased locomotor activity in APP/PS1 males on the Y maze test compared with sham-irradiated APP/PS1 males. We found no long-term effects of ^56^Fe irradiation on the locomotor activity of WT male and female mice, in agreement with studies on more short-term effects of irradiation on locomotor activity post-irradiation in mice and rats [18,27,37,38,39]. We have previously reported that in the 1–2 months post-irradiation timeframe, small but significant differences in locomotor activity manifest predominantly in male mice [26]. Unlike other low-dose HZE radiation-induced behavioral changes, these locomotor effects appear to resolve with time. However, one recent study found no effect of 0.10, 0.50, or 1 Gy ^56^Fe irradiation on locomotor activity in WT mice up to 20 months post-irradiation [40]. We also found little to no effects of irradiation on depressive-like or anxiety-like behavior in WT mice or APP/PS1 male mice, in agreement with previous studies [18,40], including our own [26]. Of note, one previous study found that male but not female mice demonstrated increased anxiety-like behavior 80 days, but not 45 days post-irradiation [36].

We found that irradiated male mice showed some long-term deficits in spatial and fear memory. APP/PS1 males in both irradiation groups trended towards lower percentages of spontaneous alternation in the Y maze test, while WT males in the 0.50 Gy group males froze significantly less in the CFC test, indicating a deficit in fear memory. We previously found no early differences between any sex, genotype, or dose groups on the Y maze test, and that APP/PS1, but not WT, mice showed early fear memory deficits on the CFC test [26]. Our past and current results do not agree with Miry et al., who found that male WT mice show deficits in the Active Avoidance test 2 months post ^56^Fe irradiation, but that the deficits resolve by 12 months post-irradiation [40]. However, in a different study, we found that 0.10 Gy ^56^Fe irradiation did not impair fear memory in male APP/PS1 mice 6 months after irradiation, though both males and females showed impairment on the novel object recognition test [20]. Of note, one recent study found that fractionated ^56^Fe improved hippocampal-dependent memory but impaired striatal-dependent memory in female WT mice [41].

^56^Fe is one of many heavy ions that astronauts will encounter in space, and sex differences have been reported with other ion species as well. For example, ^28^Si radiation has been shown to prevent the survival of new neurons in male, but not female, mice [34], while ^4^He has been shown to cause cognitive deficits and increases in neuroinflammation in male mice [42]. More recent studies have increasingly taken advantage of newly available multi-ion irradiation paradigms that more closely mimic the space environment, such as Galactic Cosmic Radiation (GCRsim). Studies to date have found that GCRsim reduced the firing frequency of neuronal action potentials and impaired hippocampus-dependent memory and hippocampal LTP in mice [43,44], and decreased social interaction, increased anxiety-like behaviors, and impaired social recognition in male mice, corresponding with synaptic loss in the hippocampus [36]. Our behavioral results with ^56^Fe, which is a component of GCRsim, largely support these findings. Combined with previous research, our results suggest that deficits in memory are governed by a complex interplay of age-, sex-, radiation-, and genotype-dependent factors, as well as yet to be determined compensatory mechanisms.

The present study is one of the first to examine the long-term effects of ^56^Fe radiation on startle and PPI responses. We found that ^56^Fe irradiation reduced the startle response in male WT mice (90 and 120 dB) and increased it in female APP/PS1 mice (110 and 120 dB), while 0.10 Gy ^56^Fe radiation decreased PPI in APP/PS1 males compared to those in the 0.50 Gy group. Dopamine plays an important role in PPI and startle response [45], and as such our results suggest a long-term perturbation of dopaminergic pathways. Multiple studies in rats have found that ^56^Fe irradiation disrupts the development of Conditioned Taste Avoidance produced by an injection of amphetamine [46,47,48]. However, a study in mice found that 5 Gy ^56^Fe inhibited the startle response at 4–5 days following irradiation but had no effects on PPI and concluded that HZE radiation disrupts the dopaminergic system, but probably not enough to affect PPI in the short term [49]. Taken together, our data suggest that radiation-induced changes to the dopaminergic system persist months after exposure in a sex- and genotype-specific manner.

In addition to altered dopaminergic function, several other factors may contribute to the impacts of space radiation on cognition. The early effects of radiation on synaptic function are widely reported [14,50,51], and include decreased synaptic density [52], decreased readily releasable vesicular pools and expression of NMDA receptor subunits [14] and decreased expression of presynaptic proteins, which correlate with learning and memory deficits [13]. One recent study showed that neurogenesis in the dentate gyrus of the hippocampus was suppressed 2 months after ^56^Fe irradiation, but rebounded 10 months later, suggesting that compensatory mechanisms may mitigate irradiation-induced damage [40]. These synaptic changes may be caused by ^56^Fe-mediated reactive oxygen species overload, which has been shown to cause neuronal death in the hippocampus and subsequent cognitive deficits as early as 24 h post-irradiation [6]. These factors could also explain the radiation-induced deficits in fear-based and spatial memory in the present study.

Despite our behavioral results, we found no differences between groups in the integrated density of synaptophysin staining of the CA1, CA3, and prefrontal cortex regions 8 months following irradiation. Nor were there any effects of ^56^Fe irradiation on hippocampal and cortical PSD-95 or MAP2 integrated densities. However, it cannot be ruled out that the whole brain Western blots were not sensitive enough to capture synaptic changes. Of note, one study using rats found that hippocampal- and prefrontal-cortex-dependent cognitive domains are not necessarily concomitantly impaired by 2 Gy ^56^Fe irradiation [53], suggesting that multiple behavioral testing paradigms are necessary to demonstrate the extent of ^56^Fe radiation-induced cognitive impairment. While the exact effects of ^56^Fe irradiation on hippocampal memory are yet to be elucidated, our present results suggest that radiation-induced hippocampal memory impairment may be sex- and genotype dependent.

We previously found that 1 month after irradiation, 0.50 Gy ^56^Fe irradiation decreased insoluble Aβx-42 in APP/PS1 females, and that both 0.10 and 0.50 Gy of irradiation decreased insoluble Aβx-40 in APP/PS1 females. In the present study, we found that while sham APP/PS1 females had significantly higher insoluble Aβx-40 and Aβx-42 than sham APP/PS1 males, exposure to 0.50 Gy ^56^Fe irradiation significantly increased Aβx-40 in males but not in females. Similarly, 0.50 Gy ^56^Fe irradiation increased hippocampal R1282 staining in APP/PS1 males but not APP/PS1 females, even though sham-irradiated APP/PS1 females had significantly higher hippocampal R1282 staining than sham-irradiated APP/PS1 males. These results agree with those in our other previous study, which showed that ^56^Fe irradiation accelerated Aβ plaque pathology in male APP/PS1 mice 6 months later [20]. Taken together, these results suggest that ^56^Fe irradiation has a long-lasting effect on plaque pathology in male, but not female, APP/PS1 mice.

The present study found that ^56^Fe irradiation increased gliosis in male and female APP/PS1 mice. Interestingly, our previous study showed that ^56^Fe irradiation decreased gliosis and Aβ pathology in APP/PS1 females, with no such changes in APP/PS1 males. Along with our present results, this suggests that while ^56^Fe irradiation may attenuate harmful Aβ pathology and gliosis in APP/PS1 females 1 month following irradiation, the effect on gliosis is transient. Moreover, while APP/PS1 males may not show deleterious Aβ pathology or gliosis 1 month following irradiation, at 8 months following irradiation Aβ pathology and gliosis increases in a radiation dose-dependent manner. The relationship between radiation and gliosis remains unclear. Other studies have found that single doses of ^56^Fe irradiation increased the number of activated microglia in mice [11,36]. ^56^Fe irradiation did not seem to affect the number of microhemorrhages 8 months later, even though we had previously found that ^56^Fe irradiation increased the number of microhemorrhages in WT males in a dose-dependent manner shortly after ^56^Fe exposure [26]. 

The sex differences regarding Aβ pathology and gliosis were echoed in our PET imaging with the TSPO ligand PET tracer ^18^F-GE180. We found that hippocampal uptake of ^18^F-GE180 was increased in APP/PS1 male, but not APP/PS1 female, mice following irradiation. However, our immunohistochemical and PET imaging results are limited because the PET imaging was undertaken before and after irradiation with only 0 and 0.50 Gy dose groups, whereas immunohistochemistry was conducted only after irradiation with 0, 10, and 0.50 Gy dose groups. Our previous study also found little to no effects of ^56^Fe irradiation on WT mice but found that 0.50 Gy ^56^Fe irradiation decreased tracer uptake in APP/PS1 females [26], possibly related to the reduced amyloid load in brain.

Similarly, we found that ^56^Fe irradiation altered plasma levels of IL-10, IL-4, and TNFα in APP/PS1 mice. IL-4 has been previously shown to induce neuroprotective microglial phenotypes in vitro and protect against ischemic damage in vivo [54]. Thus, the reduction in IL-4 in the present study suggests that APP/PS1 females may have impaired protective responses to irradiation. Accordingly, we found that female APP/PS1 mice in the 0.50 Gy group trended towards having significantly more Iba-1 positive staining than sham-irradiated APP/PS1 females. Male APP/PS1 mice in the 0.10 Gy group had reduced IL-10 compared with both the sham and 0.50 Gy groups. IL-10 is a classic anti-inflammatory cytokine [55] that may encourage neuroprotective microglial phenotypes [56]. However, we did not observe any correlations with changes in Iba-1 or CD68 staining in APP/PS1 males in the 0.10 Gy group. Other studies have shown that cytokines have biphasic responses to irradiation, suggesting that the observed changes in the present study may not be representative of the entire time course of cytokine levels following irradiation [57,58].

The present study is one of the first to examine the long-term effects of low-dose ^56^Fe irradiation on cognition, Alzheimer’s amyloid pathology, neuroinflammation, and synaptic markers on male and female, WT and APP/PS1 mice. We report that APP/PS1 male mice demonstrated generally increased inflammation and increased plaque loads 8 months following irradiation, while female APP/PS1 mice had lower survival rates at the 0.50 Gy dose. However, the female APP/PS1 mice who survived seemed less affected by ^56^Fe irradiation despite displaying more severe genotype effects. ^56^Fe irradiation had little to no effects on general health, body weight, and levels of inflammatory cytokines and synaptic markers. Our results, along with those in our earlier study, suggest that the physiological and corresponding behavioral effects of ^56^Fe irradiation are related to sex and genetic predisposition to Alzheimer’s disease, and that long- and short-term physiological and cognitive effects are not the same. Altogether, these findings in a mouse model of AD inform about possible risks associated with HZE radiation exposure that may be of relevance for astronauts on prolonged trips beyond low Earth orbit. Based on the observed interaction between radiation and disease, common disease-associated genetic variants likely to occur in the astronaut population must be considered in the context of radiation risk. Additional studies with radiation doses and qualities that more closely mimic the deep space environment are clearly required.

## 4. Materials and Methods

### 4.1. Mice

APPswe/PS1dE9 transgenic (APP/PS1) or age- and sex-matched C57BL/6J wild-type (WT) littermates were bred and aged at Brigham and Women’s Hospital (BWH) (Boston, MA, USA). APPswe/PS1dE9 mice have the Swedish APP^K594N/M595L^ human transgene as well as the PS1dE9 human transgene, both of which are under a mouse prion promotor [59]. APPswe/PS1dE9 mice exhibit AD pathology including extracellular amyloid deposits in the prefrontal cortex and hippocampus by 5–6 months of age, Aβ plaque deposition, microhemorrhages, gliosis, and cognitive deficits by 7–8 months of age [60,61,62]. Mice were transferred to Brookhaven National Laboratory (BNL) in Upton, NY, USA irradiated at 4 months of age, and returned to BWH where they underwent quarantine before being moved back into the mouse colony. Mice underwent behavioral testing at 11 months of age and were euthanized by CO_2_ exposure at 12 months of age. At both BWH and BNL, mice were housed at a constant temperature on a 12 h light/dark cycle and had ad libitum access to food (PicoLab Rodent Diet #5053) and water. All animal use was approved by the Harvard Medical School Office for Research Subject Protection—Harvard Medical Area Standing Committee on Animals and the Brookhaven National Laboratory Institutional Animal Care and Use Committee.

### 4.2. ^56^Fe Irradiation of Mice

A graphic of the experimental timeline can be found in Appendix A. In October 2015, all mice were shipped to BNL, allowed to acclimate for 3–5 days, and transferred to the NASA Space Radiation Laboratory (NSRL). For irradiation, 10 mice at a time were loaded individually into ventilated 50 mL conical tubes. The tubes with the mice were loaded into foam holders and carried into the “cave,” where the mice received whole-body irradiation from the side. The irradiation consisted of 1 GeV/n ^56^Fe (151.4 KeV/micron) at either 10 or 0.50 Gy at a rate of 20 Gy/minute. Particle fluence was 2,110,000 ions/cm^2^ for the 0.50 Gy exposures and 422,000 ions/cm^2^ for the 0.10 Gy exposures. Following a Poisson distribution, this equates to an estimated average 6.33 and 1.27 particle traversals through a circular target 20 μm in diameter with un-hit fractions (i.e., the probability of a target receiving no traversals) of 0.002 and 0.282, respectively. There were 13–16 mice per sex/genotype/dose. Following irradiation, mice were returned to their cages. Sham-irradiated mice were put into the 50 mL ventilated conical tubes, loaded into the foam holders and carried around for an equivalent time to that of the irradiated mice, but were not taken into the cave.

### 4.3. Behavioral Tests

Six-to-nine mice per group underwent various behavioral tests (Table 1) 7 months after irradiation, at 11 months of age. The SmithKline Beecham, Harwell, Imperial College, Royal London Hospital, phenotype assessment (SHIRPA) was used to measure baseline health and general function in mice.

#### 4.3.1. Open Field (OF)

The OF test measures locomotor activity, anxiety, and context habituation. Mice were placed in the center of a 27 cm × 27 cm test chamber and allowed to explore for 1 h. To measure locomotor activity, a computer-assisted infrared tracking system computed the total distance traveled (cm) in 5 min time bins and counts for vertical exploration. Anxiety-like behavior was measured by the amount of time spent and distance traveled in the center of the OF.

#### 4.3.2. Y Maze Spontaneous Alternation Test

The Y maze test assesses spatial memory and locomotor activity by measuring the total number of arm entries and the total distance traveled. Mice were placed in the center of the maze and allowed to explore for 6 min, and the number and sequence of arm entries were recorded.

#### 4.3.3. Rotarod Test

The rotarod test measures sensorimotor coordination and fatigue resistance. Mice underwent a habituation session at 4 rpm for 5 min to acclimate mice to the apparatus and then were tested twice (test 1 and test 2) to assess motor coordination and learning. The inter-trial interval was 3 h. For the tests, the acceleration rate was 4–40 rpm in 3 min and the latency to fall was recorded automatically.

#### 4.3.4. Elevated Plus Maze (EPM)

The EPM measures locomotor activity and anxiety-like behavior behavior. The EPM apparatus consisted of 4 arms, two open and two closed, extending from a central platform. Each arm was 30 cm in length and 5 cm in width, and the maze was 90 cm above the floor. The test was conducted in dim ambient lighting. Mice were placed on the central platform of the maze and facing an open arm. They were allowed to explore the maze for 5 min. A computer-assisted video tracking system (TopScan Software, Version 1, CleverSys Inc., Reston, VA, USA) recorded the number of open and closed arm entries and the total time spent in the open, closed, and center areas. The number of closed arm entries was considered a measure of locomotor activity, while the percent of open arm entries was used as a meas.

#### 4.3.5. Tail Suspension Test (TST)

The TST is used as a measure of depressive-like behavior. Mice were suspended by their tails for 6 min. The testing apparatus is described here [26]. Briefly, the mouse’s activity was represented as a voltage output, and the time below threshold was an estimate of how much time the mouse spends immobile, which is an indicator of depressive-like behavior.

#### 4.3.6. Contextual Fear Conditioning Test (CFC)

The CFC assesses hippocampus/amygdala-dependent learning in mice. During the training phase on day 1, mice were placed in the conditioning chamber and allowed to acclimate for 2 min. They then received 2, 2 s, 0.5 mA foot shocks (contextual conditioning) with an inter-trial interval of 2 min. In the test phase on day 2, mice were placed in the conditioning chamber but did not receive a shock. Mice were videotaped and computer vision software (TopScan Software, Version 1, CleverSys Inc., Reston, VA, USA) was used to score freezing.

#### 4.3.7. Startle Test

The acoustic startle reflex assesses sensorimotor reactivity, attention, and/or emotional state. Mice were placed in restrictive holders (Med Associates, St. Albans, VT, USA, 3.2 cm in diameter) with their heads facing the speaker. The restrictive holders were placed in individual acoustic chambers (Med Associates, St. Albans, VT, USA) interior 50.8 cm × 33 cm × 30.5 cm) that were on top of a transducer platform that measured the mouse’s response to the stimulus. After a 5 min acclimation period without white noise, mice underwent sessions of 10 blocks of 11 trials of acoustic stimuli each, for a total of 110 trials. Each acoustic stimulus lasted for 40 ms with a variable inter-trial interval lasting a variable 10–20 s, with a mean of 15 s. Stimuli ranged from 20 to 120 decibels and were presented in random order. The mouse’s physical response was recorded for 150 ms following each stimulus and sampled every ms.

#### 4.3.8. Prepulse Inhibition (PPI)

PPI assesses the extent to which a weaker stimulus reduces the startle response to a subsequent stronger stimulus and thus is a measure of sensorimotor reactivity and gating. Mice were placed in a PPI chamber (Med Associates, interior 50.8 cm × 33 cm × 30.5 cm) for a 5 min acclimation session of 70 dB white noise. The test session started immediately after the acclimation session and consisted of a habituation block of 6 presentations of startle stimuli and then 10 PPI blocks of 6 different types of trials, for a total of 60 trials. The 6 trial types are as follows: null (no stimuli), startle (120 dB), startle preceded by a prepulse of 5000 Hz 65 dB, 75 dB, or 85 dB, or prepulse alone (85 dB 5000 Hz tone). The six different trial types were presented in a random order. Each trial began with recording of the mouse’s baseline movements during a 50 ms null period. Then, there was a 20 ms prepulse tone and the mouse’s response was recorded. The startle stimuli were presented for 40 ms and 100 ms after the prepulse tone. The mouse’s response was recorded for every ms for 140 ms. after startle onset. The inter-trial interval ranged from 10 to 20 s, with an average of 15 s.

#### 4.3.9. Grip Strength (GS) Test

The GS test is a measure of forepaw grip strength. After acclimating to the room for 15 min, mice were picked up and allowed to grab the grip strength meter with their forepaws. Once both paws were on the meter, mice were pulled slowly back by their tails until they let go, and the force was measured by a force transducer. Mice were given 5 consecutive trials with a 10 s inter-trial interval in a holding cage. The adjusted average grip strength was calculated after eliminating the lowest and highest force measurements.

#### 4.3.10. Wire Hanging (WH) Test

The WH test is a measure of endurance. The apparatus consisted of a horizontal wire placed 50 cm on top of a cage with 4 cm of bedding. The wire was secured to support posts with tape. Laminated paper discs were used to ensure that mice could not climb down the support posts. After acclimating to the room for 15 m, mice were given 3 consecutive trials (maximum 3 min per trial) with 30 s inter-trial intervals in a holding cage. Mice were placed on the top of the wire, and once they gripped the wire with all 4 paws, they were gently flipped so that they were hanging upside down. The latency to fall was recorded and averaged across the 3 trials.

### 4.4. MicroPET Imaging

A subset of mice receiving sham or 0.50 Gy ^56^Fe irradiation (*n* = 4/group) underwent microPET imaging using the ^18^F-GE180 tracer 2 weeks prior to irradiation at 3.5 months of age and again 7.5 months after irradiation at 11.5 months of age to assess neuroinflammation. A more detailed description of microPET imaging can be found in our previous publication [26]. ^18^F-GE180 was synthesized on the FASTlab^TM^ synthesizer using a previously published method [63]. Briefly, ^18^O enriched H_2_O (97% enrichment) was irradiated with protons to generate ^18^F. The ^18^F anion was then reacted with the precursor molecule (GE Healthcare, Franklin Lakes, NJ, USA), resulting in the formation of ^18^F-GE180. Mice were anesthetized with 3% isoflurane (Baxter Medica, AB, Kista, Sweden) and 1 L/min of medical-grade oxygen. Following the scan, mice were injected in the tail vein with the same dose per gram of bodyweight (1.75 uCi/g) of ^18^F-GE180 solution, and then flushed with 0.1 mL of saline. For the next 60 min, mice underwent dynamic microPET imaging using a small animal PET/CT scanner (eXplore Vista, GE Healthcare, Chalfont St. Giles, UK). The spatial resolution was 1.6 mm at the center of the PET imaging field of view. For quantification, hippocampus-specific uptake and binding of the ^18^F-GE180 PET tracer, PET/CT images were registered to the MRI-based brain atlas in VivoQuant^TM^ (Invicro, Needham, MA, USA).

### 4.5. Euthanasia and Tissue Preparation

Mice were euthanized with CO_2_ inhalation and transcardially perfused with 20 mL of Phosphate buffered saline. Brains were extracted and divided sagittally. One hemibrain was fixed overnight in 4% paraformaldehyde (PFA), cryoprotected with 10% and 30% sucrose, and embedded in OCT as previously described [20]. The other hemibrain was snap-frozen in liquid nitrogen and stored at −80 °C.

### 4.6. MSD Multiplex AB Triplex-38/40/42 Protein ELISA

Brain homogenization and ELISA protocols were performed as previously described [26]. Briefly Guanidine samples were mixed overnight at 4 °C and were centrifuged at 175,000× *g* for 60 min at 4 °C. The supernatant (insoluble fraction) was transferred, aliquoted and stored at −80 °C. Cerebral levels of Aβx-38, x-40, and x-42 were measured simultaneously by the Human/Rodent 4G8 Aβ Triplex Ultra-Sensitive Assay (Meso Scale Diagnostics, Rockville, MD, USA).

### 4.7. MSD Cytokine ELISA

The Proinflammatory Panel 1 (mouse) V-PLEX ELISA kit (Meso Scale Diagnostics, Rockville, MD) was used to quantify plasma cytokine levels of IFN-γ, IL-1β, IL-2, IL-4, IL-5, IL-6, KC-GRO, IL-10, IL-12p70, and TNF-α. Plasma samples were diluted 1:2 with diluent provided by the kit, and the ELISA was conducted following the procedures given by the kit insert.

### 4.8. Immunohistochemistry, Histology, and Quantification

Immunohistochemical methods are described previously in detail [26]. Briefly, 20 μm, OCT-embedded, frozen mouse brain sections were immunolabeled using the ABC ELITE method (Vector Laboratories, Burlingame, CA, USA). The R1282 rabbit polyclonal anti-Aβ antibody (1:1000, a gift from Dr. Dennis Selkoe, Brigham and Women’s Hospital) was used to assess Aβ pathology. One percent aqueous Thioflavin S (Thioflavin S; Sigma-Aldrich, St. Louis, MO, USA) was used to visualize fibrillar amyloid in plaques and blood vessels. Gliosis was assessed using anti-Iba-1 rabbit polyclonal antibody (1:500, Fujifilm Wako Chemicals, Richmond, VA, USA), anti-CD68 rat monoclonal antibody (1:200, BD Biosciences, Franklin Lakes, NJ, USA), anti-TSPO rabbit monoclonal antibody (an activated microglial marker, 1:1000; Abcam, Waltham, MA, USA), and anti-GFAP mouse monoclonal antibody (1:1000; Sigma-Aldrich, St. Louis, MO, USA). Microhemorrhages were detected using 2% ferrocyanide (Sigma Aldrich, St. Louis, MO, USA) in 2% hydrochloric acid. Immunoreactivity of R1282, Thioflavin S, and gliosis markers was quantified by BIOQUANT image analysis (Nashville, TN, USA). The percent area occupied by R1282, Iba-1, TSPO, CD68 or GFAP labeling in the entire hippocampus (HC) and/or frontal cortex (FC) was calculated for 2 equidistant sagittal sections 300 μm apart per mouse. The threshold of detection was held constant during analyses. Thioflavin S labeling was averaged by 3 consecutive sections in the middle plane of the hemibrain. Microhemorrhages were counted and averaged over 6 sections (3 consecutive sections, 2 planes) of each mouse. *n =* 6–9 mice for each immunohistochemical or histological analysis.

### 4.9. Western Blot Analysis for Synaptic Markers

Western blot was performed as we reported previously [64]. Briefly, T-per soluble brain homogenates were separated on 12% Bis-Tris gels (Invitrogen, Waltham, MA, USA). Proteins on gels were transferred to PVDM membranes and probed with synaptophysin (1:3000, Sigma-Aldrich, St. Louis, MO, USA) or VGlut2 (1:3000, MilliporeSigma, Burlington, MA, USA), and post-synaptic markers: PSD-95 (1:2000, MilliporeSigma, Burlington, MA, USA) or Homer-1 (1:1000, R&D Systems, Minneapolis, MN, USA), with the anti-GAPDH antibody (1:3000, Abcam, Waltham, MA, USA) as a protein loading control. After blocking and incubation with primary antibodies overnight, protein bands of interest were visualized by binding of correlated immunoreactive dye-labeled secondary antibodies (Li-Cor, Lincoln, NE, USA). The intensity of the protein bands was analyzed by Odyssey imaging system (Li-Cor, Lincoln, NE, USA).

### 4.10. Statistical Analyses

All data are expressed as the mean ± SEM. A value of *p* < 0.05 was considered significant and *p* < 0.1 was considered a notable trend for all statistical tests. Behavioral data were analyzed in StatView 5.0 (SAS Institute, Cary, NC, USA) using 3-way ANOVAs for sex, genotype, and dose followed by comparisons within sex/genotype groups (i.e., between 0, 0.10, and 0.50 Gy of one sex/genotype combination) and between same-sex nonirradiated controls using Fisher’s Protected Least Significant Difference (PLSD). Non-behavioral data were analyzed in Prism 8.0 (GraphPad, San Diego, CA, USA) following a similar scheme but with Tukey’s multiple comparison corrections for comparisons within sex/genotype groups. Sham-irradiated groups (male WT vs. male APP/PS1, female WT vs. female APP/PS1, female WT vs. male WT, female APP/PS1 vs. male APP/PS1) were compared with two-tailed, unpaired *t* tests or Mann–Whitney U tests if the data were not normally distributed. Survival curves were made and analyzed using Prism version 8.0′s (Graphpad, San Diego, CA, USA) log-rank (Mantel Cox) test. Bodyweights were analyzed using a 2-way repeated-measures ANOVA. PET imaging data were analyzed by a 2-way ANOVA followed by Bonferroni’s post hoc test. Data that failed normality (Anderson–Darling, D’Agostino–Pearson omnibus, and Shapiro–Wilk tests) or homoscedasticity (Spearman’s test) assumptions were analyzed with multiple, fewer-dimensional ANOVAs (2-way within sex followed by 1-way within sex/genotype groups if needed), and corrections for non-normal distributions were applied as appropriate.

## Figures and Tables

**Figure 1 ijms-22-13305-f001:**
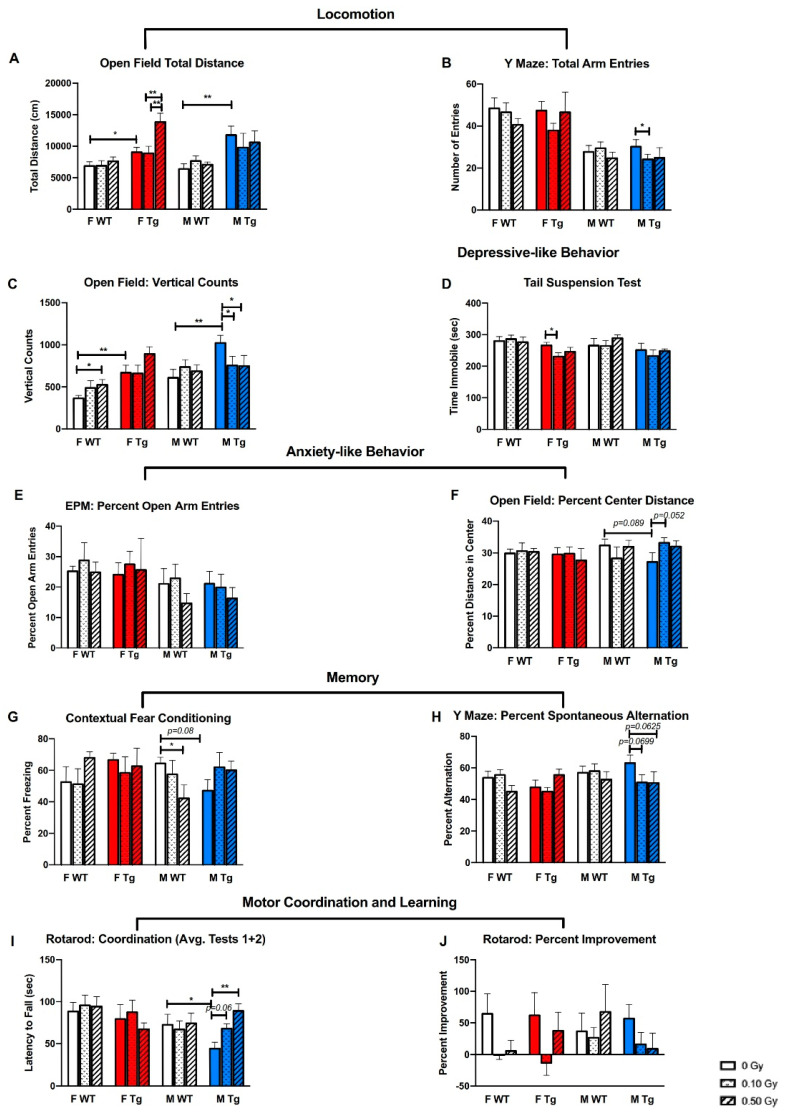
Late behavioral changes in APP/PS1dE9 and WT mice after a single dose of whole-body ^56^Fe irradiation were dependent on sex, genotype, and dose. Four-month-old male and female APP/PS1dE9 APP/PS1 (Tg) mice and WT littermates were irradiated with 0, 0.10 or 0.50 Gy of ^56^Fe irradiation at BNL and a subset of these mice (*n* = 4–9 mice/group) went through behavioral tests starting at seven months post-irradiation. (**A**–**C**) ^56^Fe irradiation had sex- and genotype-specific effects on tests of locomotor activity, including total distance in the open field (**A**), total arm entries in the Y maze (**B**), and total vertical counts in the open field (**C**). (**D**–**F**) ^56^Fe irradiation had sex- and genotype-specific effects on tests of depressive-like behavior including less time immobile on the tail suspension test in female APP/PS1 mice in the 0.10 Gy group (**D**), but not anxiety-like behaviors, such as the percent entries on the open arm of the elevated plus maze (**E**), and the percent distance traveled in the center of the open field (**F**). (**G**,**H**) ^56^Fe irradiation impaired fear memory in male WT mice on the contextual fear conditioning test (**G**), and there were trends towards ^56^Fe irradiation affecting spatial memory in male APP/PS1 mice in the Y maze test (**H**). (**I**,**J**) ^56^Fe irradiation had late effects on motor coordination of male mice (**I**) but not on motor learning (**J**). *: *p* < 0.05, **: *p* < 0.01. Data were analyzed with 3-way ANOVAs followed by 1-way ANOVAs of male and female mice with Fisher’s Protected Least Significant Difference.

**Figure 2 ijms-22-13305-f002:**
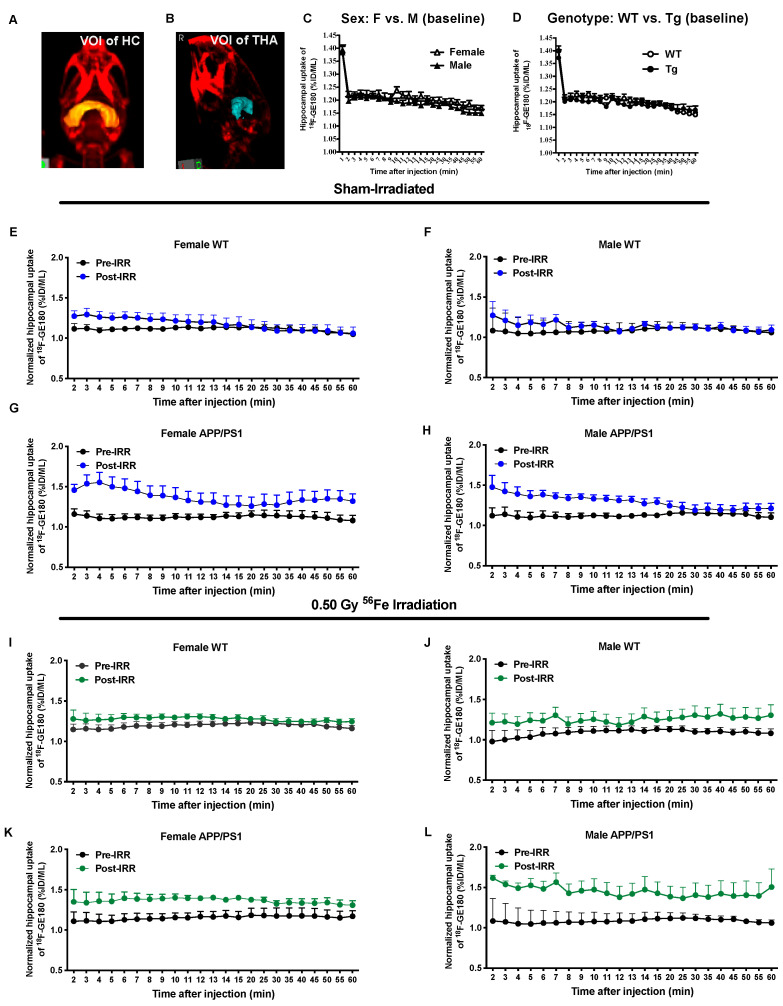
Hippocampal uptake of ^18^F-GE180 PET tracer was increased in APP/PS1 mice without ^56^Fe irradiation. (**A**,**B**) For PET imaging analysis, volume of interest (VOI) for hippocampus (HC, **A**) and the reference region, thalamus (THA, **B**), were as shown. (**C**,**D**) There was no significant sex (**C**) or genotype (**D**) difference in baseline uptake (pre-irradiation PET at 3.5 months of age) of ^18^F-GE180. (**E**–**H**) APP/PS1 female (**G**) and male (**F**) mice showed pathology- and aging-associated increases in post-irradiation hippocampal uptake of ^18^F-GE180 when compared with female and male WT mice (**E**,**F**). (**I**–**L**) The 0.50 Gy ^56^Fe irradiation increased post-irradiation hippocampal uptake of ^18^F-GE180 at 11 months of age in APP/PS1 female (**K**) and male (**L**) mice compared to female and male WT mice (**I**,**J**). *n* = 3–4 mice/group. A 2-way ANOVA followed by Bonferroni’s post hoc test was used for ^18^F-GE180 uptake analysis.

**Figure 3 ijms-22-13305-f003:**
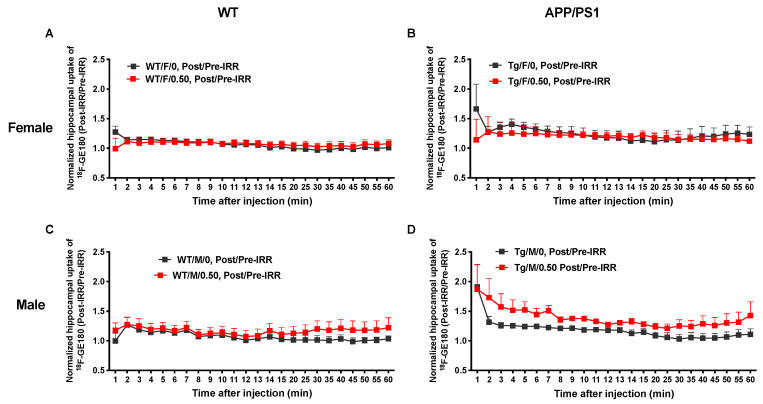
^56^Fe irradiation increased hippocampal neuroinflammation in male APP/PS1 mice. To investigate radiation-specific effects on neuroinflammation, we normalized post-irradiation ^18^F-GE180 PET signal by its pre-irradiation ^18^F-GE180 PET signal for each group (post-irradiation/pre-irradiation). ^56^Fe irradiation did not increase neuroinflammation (^18^F-GE180 uptake) in female mice (**A**,**B**) or in WT males (**C**). However, 0.50 Gy of ^56^Fe significantly increased neuroinflammation in APP/PS1 males (**D**). *n* = 3–4 mice/group. Data were analyzed with a 2-way ANOVA followed by Bonferroni’s post hoc test.

**Figure 4 ijms-22-13305-f004:**
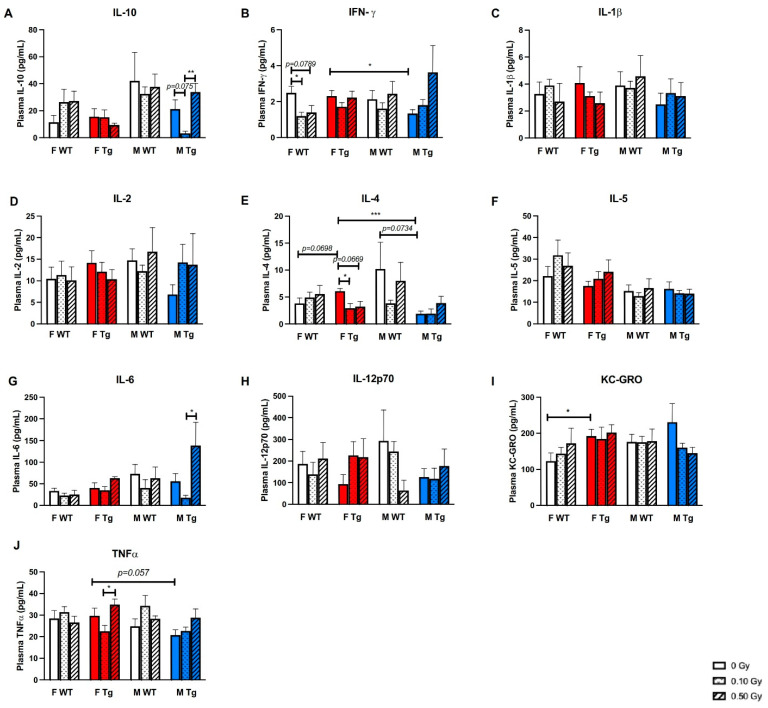
Low-dose ^56^Fe irradiation lowered plasma levels of IL-10 in male APP/PS1 mice compared with sham-irradiation controls. a–j Plasma levels of 10 cytokines, IL-10 (**A**), INF-γ (**B**), IL-1β (**C**), IL-2 (**D**), IL-4 (**E**), IL-5 (**F**), IL-6 (**G**), IL-12p70 (**H**), KC-GRO (**I**), and TNF-α (**J**), were measured by MSD ELISA. No significant late effects of ^56^Fe irradiation on plasma cytokines were observed, except low-dose irradiation resulted in an overt reduction in IL-10 in male APP/PS1 mice and IL-4 in APP/PS1 female mice versus sham-irradiated male and female APP/PS1 controls (**A**,**E**), as well as a reduction in IFN-γ in WT females (**B**). We saw baseline genotype effects on levels of KC-GRO (**I**) and baseline sex effects in APP/PS1 shams on levels of IFN-γ (**B**) and IL-4 (**E**) *n* = 5–7 mice/group. * *p* < 0.05, ** *p* < 0.01, *** *p* < 0.001. Data were analyzed by 3-way ANOVAs followed up with 1-way ANOVAs with Tukey comparisons and planned, unpaired, 2-tailed t tests between sham groups as necessary. Non-parametric data were analyzed with the Kruskal–Wallis and/or Mann–Whitney U tests.

**Figure 5 ijms-22-13305-f005:**
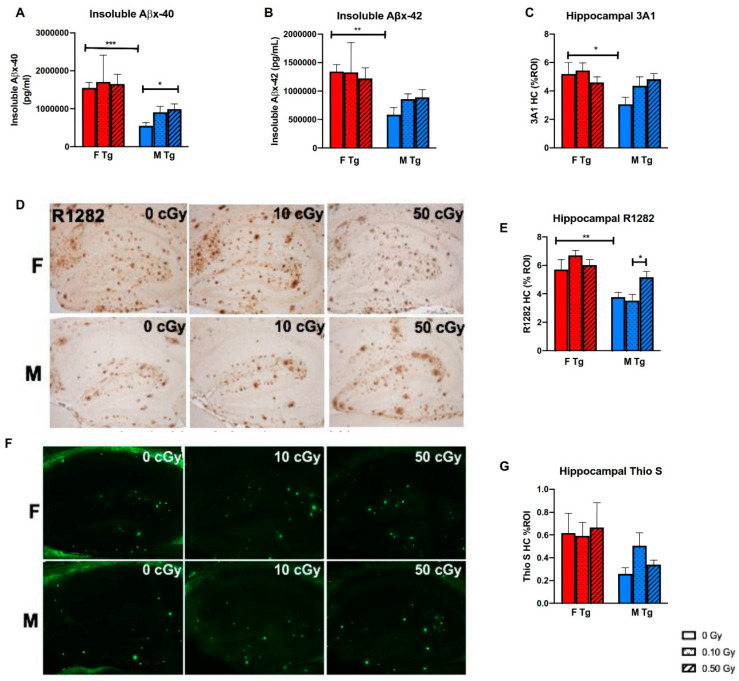
^56^Fe irradiation caused elevated cerebral Aβ levels in APP/PS1 males, but not in APP/PS1 females, at 8 months post-irradiation. (**A**,**B**) Insoluble Aβx-40 (**A**) and Aβx-42 (**B**) levels were quantified by an MSD 4G8 Aβ-triplex ELISA (*n* = 9–16 mice/group). (**E**–**G**) Amyloid pathologies were assessed by R1282 immunohistochemical (IHC) staining (**E**) and were quantified by the % region of interest (%ROI) in the brain regions of hippocampus (**D**) using the BioQuant program (*n* = 6–9 mice/group), and were further confirmed by 3A1 immunoreactivity (**C**). Fibrillar Aβ was measured by Thioflavin S staining (**G**), and % ROI was quantified in the hippocampus (**F**) using ImageJ (*n* = 6–9 mice/group). * *p* < 0.05, ** *p* < 0.01, and *** *p* < 0.001. Data were analyzed by 2-way ANOVAs followed up with 1-way ANOVAs with Tukey comparisons and planned, unpaired, 2-tailed t tests between sham groups as necessary.

**Figure 6 ijms-22-13305-f006:**
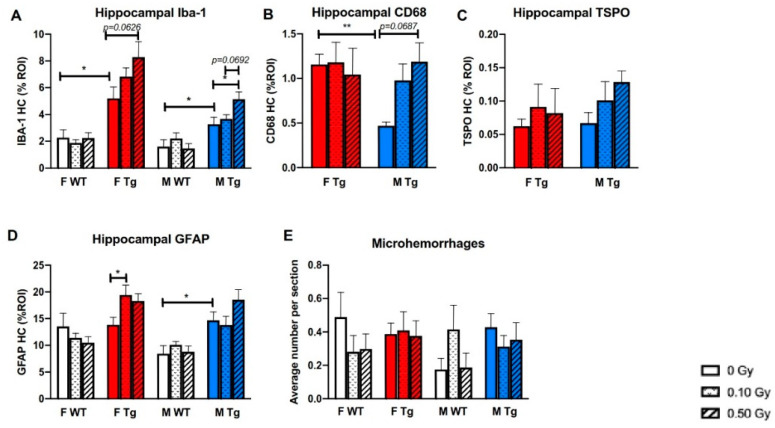
^56^Fe irradiation increased gliosis in female and male APP/PS1 mice but did not affect the number of microhemorrhages. (**A**–**D**) Late effects of ^56^Fe irradiation on hippocampal gliosis were assessed by immunohistochemical staining and % ROI quantification of microglial markers Iba-1 (**A**), CD68 (**B**) and TSPO (**C**), as well as an astrocyte marker, GFAP (**D**). Microhemorrhages were assessed by Prussian blue staining for hemosiderin deposits (**E**) (*n* = 6–9 mice/group). * *p* < 0.05 and ** *p* < 0.01. Data were analyzed by 2- or 3-way ANOVAs followed up with 1-way ANOVAs with Tukey comparisons and planned, unpaired, 2-tailed t tests between sham groups as necessary. Non-parametric data were analyzed with the Kruskal–Wallis and/or Mann–Whitney U tests.

**Table 1 ijms-22-13305-t001:** Behavioral tests.

Behavioral Measures	Tests
General Health	SHIRPA
Grip and Muscle Strength Endurance	Grip Strength (GS) Wire Hanging (WH)
Depression	Tail Suspension Test (TST)
Anxiety	Open Field (OF) Elevated Plus Maze (EPM)
Locomotor Activity	Open Field (OF) Y Maze (YM) Elevated Plus Maze (EPM)
Motor Coordination	Rotarod
Motor Learning	Rotarod
Short-Term Memory	Y Maze (YM)
Fear Memory	Contextual Fear Conditioning (CFC)
Sensory Reactivity	Startle
Sensory Gating	Pre-Pulse Inhibition (PPI)

## Data Availability

The datasets generated during and/or analyzed during the current study are available from the corresponding author upon reasonable request.

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
