# Peer review of "Long-Term Sex- and Genotype-Specific Effects of 56Fe Irradiation on Wild-Type and APPswe/PS1dE9 Transgenic Mice"

_ijms, 2021, doi:10.3390/ijms222413305_

Round 1

Reviewer 1 Report

Comments to authors:
The authors address the important and challenging issue long term sex- and genotype-specific effects of 56Fe irradiation, which is a component of space radiation. The main finding of this article, highlighting the sex differences in radioresistance of the central nervous system (CNS), is thought-provoking. Although many technical limitations remain, the potential significance of this research for human activities in space, such as long-term flights, cannot be ignored. This reviewer believes that the article submitted for review will be of interest to both specialists in this field and similar specialties and can be recommended for publication in IJMS.

One of possible technical issue is why X-ray-based analysis is not used as a comparison. Since iron ion exposure is a very specific exposure condition, it is not possible to ensure reproducibility by other researchers. For example, by adding and comparing the results of X-ray exposure conditions in this experimental system, it will be easier to compare this study with the studies of other researchers. Comparisons with other types of radiation, such as protons, as well as X-rays, will provide a more multifaceted view of the causes of gender differences in radiation tolerance. Did the authors only examine 56Fe exposure?

Second possible issue is the expression level of estrogen in female transgenic mice, because the one of underlying mechanisms of sex difference in radioresistance would be estrogen as the authors mentioned. As a phenotype of the APPswe/PS1dE9 transgenic mice, did the authors confirm that sex hormone secretion is unchanged compared to wild-type mice?

Lastly, in the article absent the Conclusions section, and a small paragraph at the end of the article, consists of meager conclusions from the study and the prospects for using the results achieved.

Minor Comments:
Page 17; line 623-624: The author should explain how to exam the hippocampus-specific uptake and binding of the 18F-GE180 PET tracer. MRI is the most common method to measure hippocampal volume, but how did the authors identify the hippocampus with micro-PET/CT? What is the spatial resolution of micro-PET/CT?

Reviewer 2 Report

Very well designed study, and very interesting data on neuroinflammation. It is excellent. FEW CENTERS CAN PROVIDE Fe 59 IRRADIATION.IT IS IMPORTANT FOR ADDING TO OUR UNDERSTANDING OF THE LETE EFFCETS OF SPACE FLIGHT,PARTICULARLY NEUROCOGNITIVE AND NEUROINFLAMMATION.
